

# Development of a quantitative loop-mediated isothermal amplification assay for the field detection of *Erysiphe necator*

Lindsey D. Thiessen[1], Tara M. Neill[2] and Walter F. Mahaffee[2]

[1] Department of Botany and Plant Pathology, Oregon State University, Corvallis, OR, USA
[2] Agricultural Research Service, United States Department of Agriculture, Corvallis, OR, USA

## ABSTRACT

Plant pathogen detection systems have been useful tools to monitor inoculum presence and initiate management schedules. More recently, a loop-mediated isothermal amplification (LAMP) assay was successfully designed for field use in the grape powdery mildew pathosystem; however, false negatives or false positives were prevalent in grower-conducted assays due to the difficulty in perceiving the magnesium pyrophosphate precipitate at low DNA concentrations. A quantitative LAMP (qLAMP) assay using a fluorescence resonance energy transfer-based probe was assessed by grape growers in the Willamette Valley of Oregon. Custom impaction spore samplers were placed at a research vineyard and six commercial vineyard locations, and were tested bi-weekly by the lab and by growers. Grower-conducted qLAMP assays used a beta-version of the Smart-DART handheld LAMP reaction devices (Diagenetix, Inc., Honolulu, HI, USA), connected to Android 4.4 enabled, Bluetooth-capable Nexus 7 tablets for output. Quantification by a quantitative PCR assay was assumed correct to compare the lab and grower qLAMP assay quantification. Growers were able to conduct and interpret qLAMP results; however, the *Erysiphe necator* inoculum quantification was unreliable using the beta-Smart-DART devices. The qLAMP assay developed was sensitive to one spore in early testing of the assay, but decreased to >20 spores by the end of the trial. The qLAMP assay is not likely a suitable management tool for grape powdery mildew due to losses in sensitivity and decreasing costs and portability for other, more reliable molecular tools.

## INTRODUCTION

Molecular techniques, such as PCR, are capable of being used to detect specific pathogens in air samples with high sensitivity and specificity (*Carisse, Bacon & Lefebvre, 2009*; *Carisse et al., 2009b*; *Falacy et al., 2007*; *Thiessen et al., 2016*; *West et al., 2008*). The detection of airborne pathogen inoculum has been improved through the development of quantitative PCR (qPCR) assays that allow for near real-time monitoring of inoculum concentration (*Carisse et al., 2009b*; *Rogers, Atkins & West, 2009*; *Temple & Johnson, 2011*; *Thiessen et al., 2016*). Despite the utility of qPCR to monitor pathogens, it is often

Corresponding author
Lindsey D. Thiessen,
ldthiess@ncsu.edu

impractical due to requirements for experienced laboratory staff and expensive equipment to accurately assess pathogen concentration (*Notomi et al., 2000*; *West et al., 2008*).

Loop-mediated isothermal amplification (LAMP) assays could be an inexpensive alternative for detection in the field or at remote facilities. LAMP can use relatively inexpensive and mobile equipment and utilizes the *Bst* polymerase that has a high tolerance to reaction inhibitors (*Kubota et al., 2011*), which allows for quick, minimal DNA extraction protocols. These traits make LAMP useful in field detection assays (*Harper, Ward & Clover, 2010*; *Kubota et al., 2008*; *Temple & Johnson, 2011*; *Tomlinson, Barker & Boonham, 2007*; *Tomlinson, Dickinson & Boonham, 2010*).

Loop-mediated isothermal amplification has been developed for monitoring inoculum in numerous plant pathosystems, including grape powdery mildew (*Erysiphe necator*), fire blight of pear (*Erwinia amylovora*), and gray mold (*Botrytis cinerea*) (*Temple & Johnson, 2011*; *Thiessen et al., 2016*; *Tomlinson, Dickinson & Boonham, 2010*). Traditional LAMP assays produce a magnesium pyrophosphate precipitate when DNA is amplified that can be detected with the human eye; however, in low concentrations of target DNA, precipitate may be difficult to observe (*Jenkins et al., 2011*; *Kubota et al., 2011*; *Thiessen et al., 2016*) or require expensive equipment (*Temple & Johnson, 2011*). Several dyes have been explored to improve detection including SYBR green (*Notomi et al., 2000*), hydroxynaphthol blue (*Cardoso et al., 2010*), and other synthetic dyes (*Fischbach et al., 2015*), but the dyes have the potential to inhibit LAMP reactions or require the use of spectrophotometers, which increase labor and equipment costs. The use of a fluorescence resonance energy transfer (FRET)-based probe, allows for specific detection of LAMP products and target quantification from field samples without inhibiting amplification (*Kubota et al., 2011*), and several portable fluorescence-reading LAMP devices have been made commercially available, such as the Genie (Optigene Ltd., West Sussex, UK) and Bioranger (Diagenetix, Inc., Honolulu, HI, USA). Using a fluorescent probe also reduces potential classification error from visual detection of LAMP products, which may improve the accuracy of pathogen detection and allow for quantification.

Grape powdery mildew, caused by *E. necator*, causes damages to grape (*Vitis vinifera* L.) wherever it is produced. This disease requires numerous applications of fungicides, which are either applied on a calendar schedule from bud break (BBCH 08) until véraison (BBCH 83) or based on disease risk models (*Carisse et al., 2009a*; *Gadoury & Pearson, 1990*; *Thomas, Gubler & Leavitt, 1994*). More recently, fungicide applications have been reduced using inoculum detection systems (*Thiessen, Neill & Mahaffee, 2017*; *Thiessen et al., 2016*); however, these systems do not provide in-field inoculum concentration for producers. Additionally, the LAMP assay that was successfully designed for field use in the grape powdery mildew pathosystem had numerous false negatives or false positives, which may have been caused by difficulty in perceiving the magnesium pyrophosphate precipitate, reducing the predictive values of the LAMP assay (*Thiessen et al., 2016*). A timely and cost-effective system that improves detection of *E. necator* inoculum throughout the growing season is needed to allow growers to accurately time fungicide applications early in the growing season and adjust application intervals based on inoculum concentration.

The purpose of this research was to develop a quantitative molecular assay for commercial implementation that could be used by growers or vineyard consultants for the detection and quantification of airborne *E. necator* inoculum. The specific objectives of this project were to (1) develop a real-time, quantitative LAMP (qLAMP) assay that was sensitive and specific to *E. necator*, and (2) test field use of a mobile, qLAMP device by growers.

## MATERIALS AND METHODS

### Sample rod preparation

Sample rods were created by cutting stainless-steel 308LSI welding rods (1.1 mm in diameter) (Weldcote Metals, Kings Mountain, NC, USA) to 36 mm lengths, then sterilized and prepared according to *Thiessen et al. (2016)*. To produce a standard curve, conidial suspensions were generated by suspending *E. necator* conidia from *V. vinifera* cv. "Chardonnay" vines in a 0.05% Tween 20 (Sigma-Aldrich, St. Louis, MO, USA) and nuclease-free water solution then pipetting the conidial suspension onto rod sets resulting in rods with 100, 1,000, or 10,000 conidia per sample. Rods with one or 10 spores were created by transferring individual spores with eyelash brush. A total of six independent spore dilution series were used to generate the standard curve for the quantitative assay. Additionally, a set of sample rods containing 500 conidia was also generated using the conidial suspension to act as a positive control for all DNA extractions and molecular reactions. The rods were air dried prior to processing.

### Quantitative LAMP assay

DNA for qLAMP analysis was extracted using a quick extraction method modified from *Thiessen et al. (2016)*. Spore rods were transferred to 2-ml screw-cap tubes containing 200 µl of 5% Chelex 100 (Sigma-Aldrich, St. Louis, MO, USA) in molecular grade, DEPC-treated water. Tubes containing rods were vortexed for 5 s then placed in boiling water for 5 min. Tubes were removed from boiling water and vortexed another 5 s. The tubes were boiled for another 5 min, and then removed and allowed to cool at room temperature for 2 min. Samples were centrifuged at 16,000$g$ for 2 min to collect the contents in the tube. Rods were aseptically removed in a laminar flow hood prior to the pellet being processed using the Chelex DNA extraction process (described below). After DNA were extracted and amplified, samples were stored at −20 °C for further analyses.

The qLAMP reaction is a modified assay from *Thiessen et al. (2016)* and *Kubota et al. (2011)*, which was optimized to generate a quantification standard curve (described above). A FRET-based probe was designed using the forward loop primer region with a FAM reporter (6-carboxyfluorescein) and a quencher strand (*Kubota et al., 2011*). Each reaction contained 14.8 µl of Isothermal Master Mix with no dye (OptiGene Ltd, West Sussex, UK), internal primers FIP EN and BIP EN (2.4 µM), external primers F3 EN and B3 EN (0.24 µM), forward loop primer FAM strand (FL-F, 0.08 µM), and Quencher strand (Q-strand, 0.08 µM) to create a 25 µl reaction (Table 1). Lab-conducted qLAMP (L-qLAMP) reactions were carried out on an ABI StepOne Plus qPCR machine

**Table 1  Primers and probes used for the detection of *Erysiphe necator* ITS region.**

| Primer/probe[a] | Nucleotide sequence (5′→3′) |
|---|---|
| qLAMP[b] | |
| FIP EN | ACCGCCACTGTCTTTAAGGGCCTTGTGGTGGCTTCGGTG |
| BIP EN | GCGTGGGCTCTACGCGTAGTAGGTTCTGGCTGATCACGAG |
| F3 EN | TCATAACACCCCCCTCAAGCTGCC |
| B3 EN | AACCTGTCAATCCGGATGAC |
| FL-F | FAM–ACGCTGAGGACCCGGATGCGAATGCGGATGCGGATGCCGAAAACTGCGACGAGCCCC |
| Q-strand | TCGGCATCCGCATCCGCATTCGCATCCGGGTCCTCAGCGT–BHQ |
| qPCR[c] | |
| Unc144 forward | CCGCCAGAGACCTCATCCAA |
| Unc511 reverse | TGGCTGATCACGAGCGTCAC |
| Unc TM probe | 6FAM*-ACGTTGTCATGTAGTCTAA-MGBNFQ |

Notes:
[a] Primers and probe from qPCR assay and primers from the LAMP assay developed by *Thiessen et al. (2016)* were used to develop and test the quantitative LAMP assay.
[b] Primer concentrations in the reaction mix were 2.4 μM for FIP and BIP, 0.24 μM for F3 and B3, and 0.8 μM for Forward Loop primer FAM strand (FL-F) and Quencher strand (Q-strand). Melting temperatures for the primers were between 64 and 99 °C.
[c] Primer concentrations in the reaction mix were 400 nM for Unc144 Forward, Unc511 Reverse, and the Unc TaqMan® Probe. Melting temperatures for the primers were 59.2 and 59.9 °C, respectively.

(Applied Biosystems, Grand Island, NY, USA). Reaction conditions were 65 °C for 45 min followed by 80 °C for 5 min. All reactions were run in triplicate.

The reaction time threshold ($R_T$) values, measured in minutes, of the spore standards were averaged and used to create a log-linear standard curve against which unknown samples were compared (Fig. 1). A log-linear curve is required to describe the assay because LAMP amplification rate is faster than exponential due to concatenation of amplicon (*Mori et al., 2001*). A 500-conidia extraction control, 100 and 1,000-conidia positive controls, as well as non-template controls were included in all reaction setups. Unknowns were compared to the standard curve to determine relative spore quantity. Positive control samples were also compared to the standard curve to determine extraction efficiency and amplification efficiency. Unknown sample $R_T$ values were adjusted based on positive control $R_T$ values if the positive controls showed poor alignment to the standard curve. To test the L-qLAMP sensitivity to target DNA, 10 separate spore concentration series were created and tested for positive amplification.

## Grower quantitative LAMP assay

Growers were provided with all equipment and supplies to conduct the DNA extraction and the qLAMP reaction protocol described above. DNA extraction and qLAMP assays were conducted in any location growers deemed appropriate (i.e., office space, winery hallway, tractor barn, kitchen table). For the grower-conducted qLAMP assay (G-qLAMP), frozen aliquots of qLAMP master mix were stored in insulated cryoboxes (VWR North America, Radnor, PA, USA) at −20 °C until reactions were conducted. All reactions were conducted in beta-version Smart-DART handheld LAMP reaction devices (Diagenetix, Inc., Honolulu, HI, USA), which connected to Android 4.4 enabled,

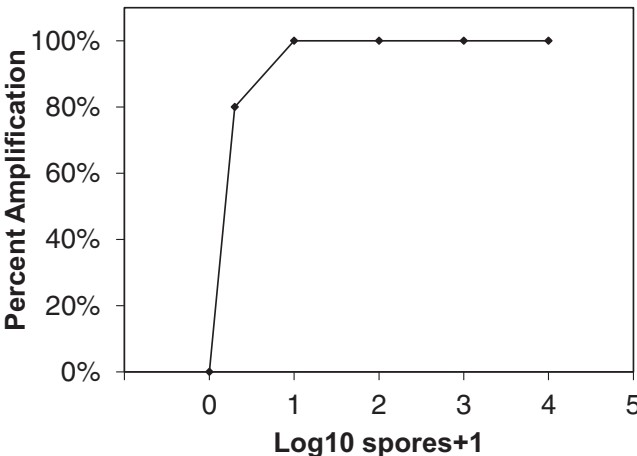

**Figure 1 Sensitivity of qLAMP assay to *Erysiphe necator* as a function of percent amplification (*y*-axis) and spore + 1 $\log_{10}$ concentrations (*x*-axis).** Each point represents the amplification of 10 separate extractions created from different *E. necator* conidia dilution series ($10^2$, $10^3$, and $10^4$ conidia concentrations), one and 10 conidia eyelash transferred spore rods, and conidia-free spore rods ($n = 10$).

Bluetooth-capable Nexus 7 tablets for output (Google, Mountain View, CA, USA). All G-qLAMP reactions were conducted in duplicate including 100-conidia positive controls and non-template controls. Reaction conditions followed the protocol described above.

Smart-DART LAMP devices provided amplification curves and the $R_T$ values associated with amplification curves. Growers were asked to determine if samples were positive, as indicated by the presence of a sigmoidal amplification curve, or negative, no amplification observed, based on the output from the handheld LAMP device.

## Quantitative PCR assay

The DNA from collected spore sampler rod pairs was extracted using the PowerSoil® DNA extraction kit (Mo Bio Laboratories, Inc., Carlsbad, CA, USA) following the manufacturer's protocols. In each set of DNA extractions, a set of positive control rods containing 500 *E. necator* conidia was included as an extraction efficiency control. *E. necator* primers developed by *Falacy et al. (2007)* were paired with a TaqMan® probe with a minor groove binder (*Thiessen et al., 2016*). All qPCR reactions contained 7.5 μl PerfeC$_T$a® qPCR ToughMix® (Quanta Biosciences, Gaithersburg, MD, USA), 400 nM final concentrations of each *E. necator* forward and reverse primers and probe (Table 1), and 1.5 μl extracted sample DNA for a 15 μl total volume. Reactions were carried out using an ABI StepOne Plus qPCR machine (Applied Biosystems, Foster City, CA, USA). All qPCR reactions were performed in triplicate, and each reaction plate contained the 500 conidia extraction control, 100 and 10,000 conidia positive reaction controls, and template-free negative control.

Cycle threshold ($C_T$) analysis was conducted using ABI StepOne™ software according to protocols by *Thiessen et al. (2016)*. Spore concentrations were determined for field samples by identifying the average $C_T$ value for each triplicate reaction, and comparing this value to the standard curve described below. Average $C_T$ values of positive controls

(100, 500, and 10,000 conidia) from each set of qPCR reactions were used to confirm the efficiency and to the suitability of the standard curve for determining conidia concentration of unknowns. The standard curve was generated by creating five independent, 10-fold conidial dilution series on the stainless-steel sampling rods 1 to $1 \times 10^5$ conidia (described above), DNA was extracted using the PowerSoil Kit (described above), and the average $C_T$ values for each conidia quantity from the five independent DNA extractions was used to fit a linear curve.

### Field sample collection and assay comparison

Custom impaction spore samplers (*Thiessen et al., 2016*), were placed at a research vineyard and six commercial vineyard locations within the Willamette Valley of Oregon. Each spore sampler contained a pair of sample rods described above. Spore samplers were run continuously, sampling 45 L/min, and sample rods were replaced daily or every Monday and Thursday (bi-weekly). Three spore samplers were placed at each of the six commercial vineyards that were collected by growers bi-weekly. The growers completely maintained one trap, processing all sample rods derived from that trap. Sample rods from the other two traps were collected by the growers and transported to the lab for processing with the L-qLAMP assay and the qPCR assay. At the Oregon State University Botany and Plant Pathology Research Farm vineyard, paired spore samplers, one for the qPCR assay and one for the qLAMP assay, were collected and processed by laboratory personnel on a daily and bi-weekly schedule.

Spore samplers for the L-qLAMP and the qPCR assays were deployed on April 15, 2013 and April 14, 2014 and sample rods were collected from bud break until véraison (BBCH 83). Spore samplers for the G-qLAMP assay were deployed April 14, 2014 and were collected until July 1, 2014. Estimates of airborne inoculum concentration derived using qPCR and qLAMP were compared to assess the accuracy of the qLAMP procedure. The G-qLAMP assay detection results were compared to the L-qLAMP assay and qPCR detection data as described below.

### Data analysis

Data was analyzed using R 3.2.1. Detections from samples collected and quantified with L-QLAMP assay were compared to qPCR assay detections using a Student's *t*-test. The G-qLAMP detection results were compared to L-qLAMP detection results using a 2 × 2 contingency table whereby the L-qLAMP results were assumed correct. Both the L-qLAMP and G-qLAMP spore detections were evaluated using a 2 × 2 contingency table whereby the qPCR assay results were assumed correct. The qLAMP assay detection accuracy, true positive proportion, true negative proportion, Fisher's exact test, and Chi-squared test were assessed comparing the qLAMP detection results to the qPCR detection results.

## RESULTS

### qLAMP assay sensitivity

The qLAMP assay showed high sensitivity to *E. necator* conidia DNA when 10 separate spore dilution series were tested (Fig. 1) with 80% of one conidia samples amplifying

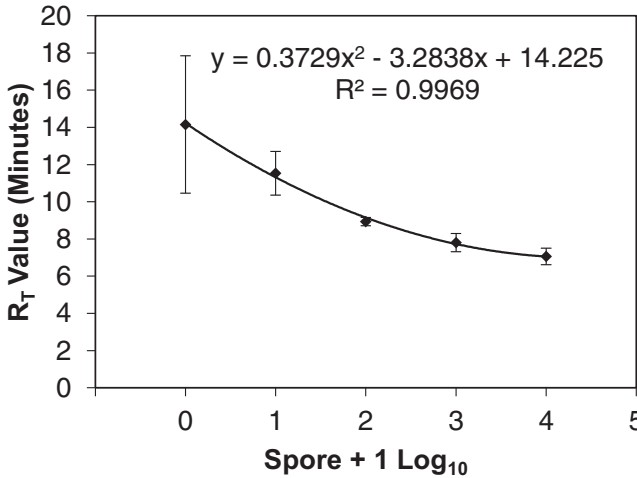

**Figure 2** qLAMP standard curve developed from six separate *Erysiphe necator* spore dilution series comparing the spore + 1 $\log_{10}$ quantity to the reaction time-threshold ($R_T$) value (minutes). The average $R_T$ value was used to determine the spore quantities of unknown samples.

using the qLAMP assay. All other spore quantities tested showed 100% amplification sensitivity within the qLAMP assay.

## qLAMP quantification

The qLAMP assay standard curve development resulted in a standard curve ($R^2 = 0.99$) when fit with a log-linear curve (Fig. 2). A log-linear curve was fit to the log spore quantity to account for the number of primers used in the assay, and the amplicon produced concatenates resulting in greater than an exponential rate of amplification. This curve was used to quantify the L-qLAMP samples collected from the Botany and Plant Pathology Research Farm vineyard. The L-qLAMP spore quantification was significantly lower than the qPCR quantification when daily samples were collected in 2013 ($P < 0.001$) (Fig. 3A), but the biweekly L-qLAMP and qPCR sample quantification was not significantly different in 2013 ($P = 0.14$) (Fig. 3B). The L-qLAMP assay significantly underrepresented spore levels for both the daily collections ($P < 0.001$) (Fig. 4A) and the biweekly collections ($P = 0.01$) (Fig. 4B) compared to the qPCR assay in 2014.

## Lab conducted qLAMP detection

Utilizing L-qLAMP for detection of *E. necator* showed similar results to qPCR assay detections in both 2013 and 2014 ($P < 0.001$) (Table 2). The L-qLAMP assay detection results were 83% and 70% accurate in 2013 and 2014, respectively compared to the qPCR assay detection results. The L-qLAMP assay detection results showed true negative proportions of 89% and 94% and true positive proportions of 78% and 37% in in 2013 and 2014, respectively. There was an unexplained loss of sensitivity in 2014 sample testing that was extensively examined (see below).

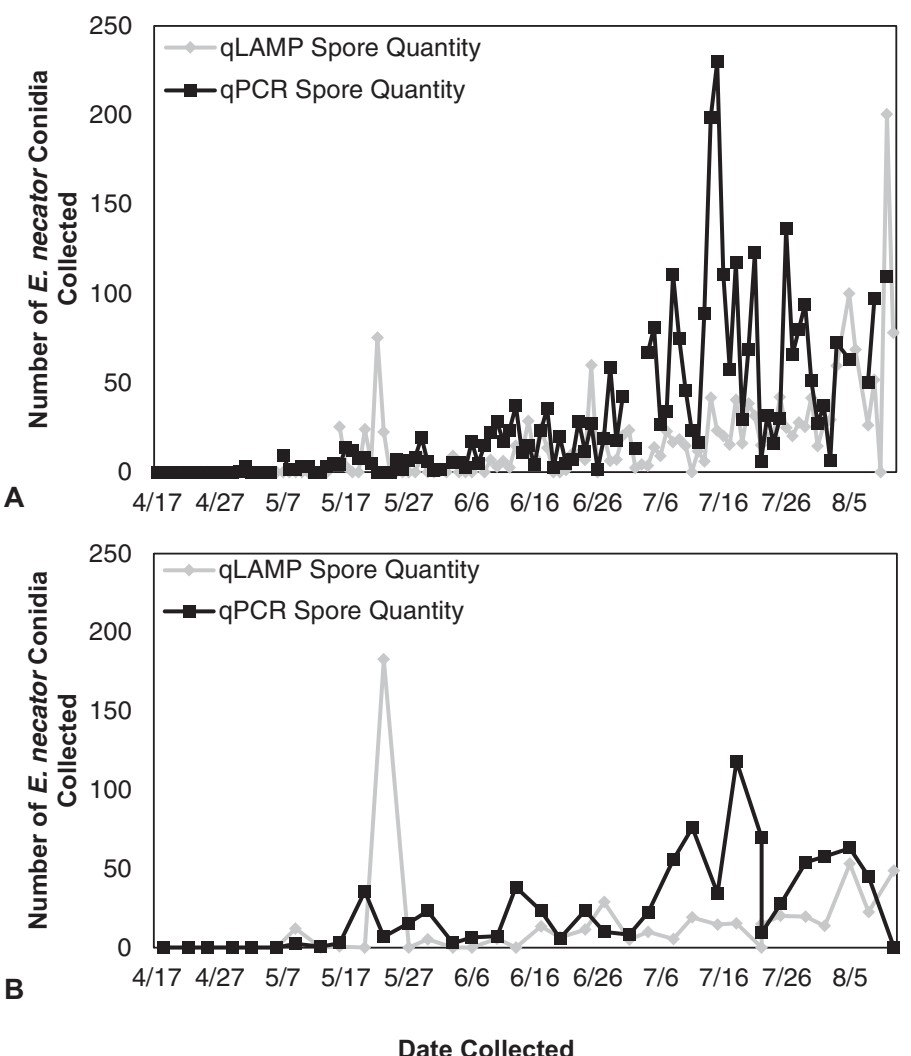

**Figure 3** *Erysiphe necator* **spore enumeration in 2013.** *Erysiphe necator* spore enumeration determined by qLAMP (gray diamond) and qPCR (black square) assays collected daily (A) and biweekly (B) from the Botany and Plant Pathology Research Farm vineyard (Corvallis, OR) during the 2013 growing season. The qLAMP spore quantification was significantly lower than the qPCR daily samples ($P <$ 0.001), but the biweekly qLAMP and qPCR sample quantification was not significantly different ($P =$ 0.14).

## Grower-conducted qLAMP assay

The software provided with the mobile LAMP device used auto-adjusting threshold values to account for noise of fluorescence readings which significantly reduced accurate quantification by growers. The G-qLAMP assay for the detection of *E. necator* was not correlated to the qPCR detection results ($P = 0.22$) (Table 2). The G-qLAMP detection results showed 82% accuracy compared to the qPCR assay results, respectively. The G-qLAMP detection results show true negative proportions of 94%, and true positive proportions of 18% compared to the qPCR detection results.

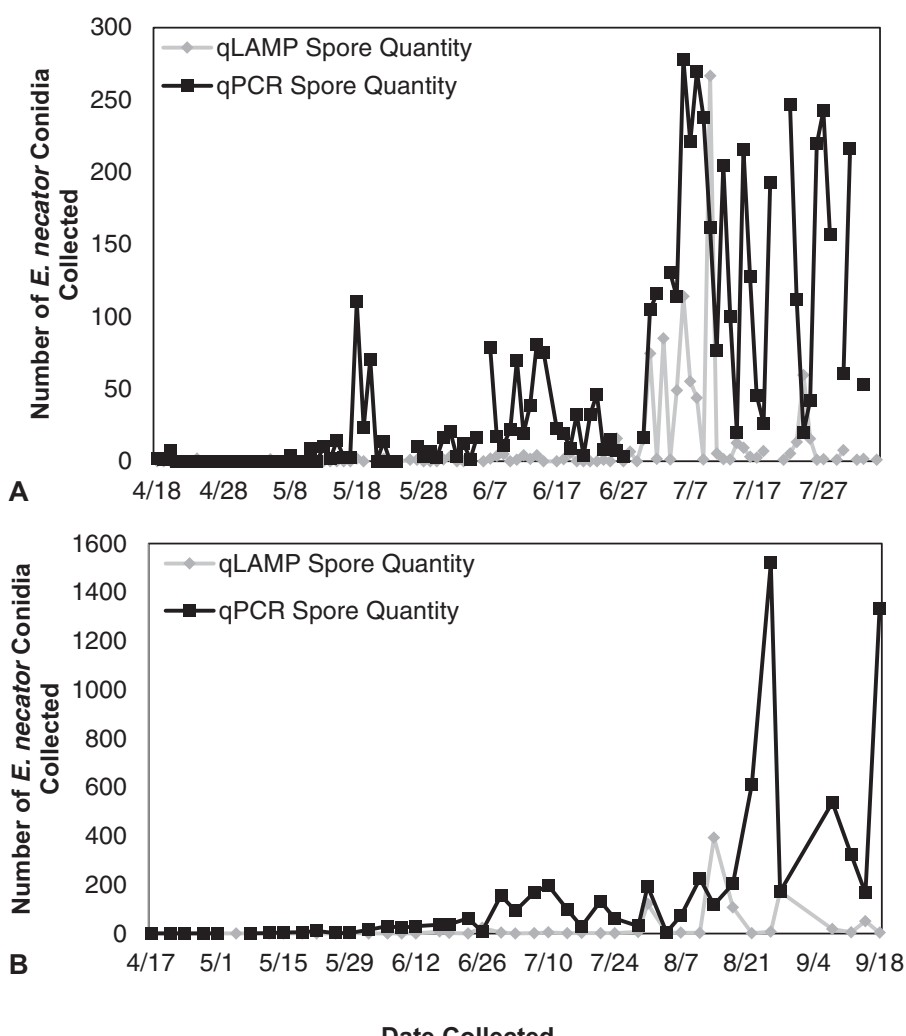

**Figure 4** *Erysiphe necator* **spore enumeration in 2014.** *Erysiphe necator* spore enumeration determined by qLAMP (gray diamond) and qPCR (black square) assays collected daily (A) and biweekly (B) from the Botany and Plant Pathology Research Farm vineyard (Corvallis, OR) during the 2014 growing season. The qLAMP assay significantly underrepresented spore levels for both the daily collections ($P < 0.001$) and the biweekly collections ($P = 0.01$) compared to the qPCR assay.

## qLAMP assay troubleshooting

Due to loss of sensitivity of the qLAMP assays to *E. necator* observed during assay testing in 2014, extensive troubleshooting was conducted. Primer purification, polymerase used (Bst; New England BioLabs, Ipswich, MA or ISO-001; OptiGene Ltd, West Sussex, UK), master mix distributer, assimilating probe removal, primer and assimilating probe manufacturer, inhibitor removal compounds in the master mix, DNA extraction and clean up, adjustment of reaction temperature, and replacement of reagents and primers were all tested. Primer purification was tested prior to the implementation of the experiment, and during the observed degradation of qLAMP sensitivity with no observable difference between reaction efficiency of HPLC or desalted primers.

**Table 2 Contingency table representing the lab quantitative LAMP assay and grower quantitative LAMP assay compared to quantitative PCR (qPCR) detection results for the presence of *Erysiphe necator* sampled from custom made impaction spore samplers from both commercial vineyards and research plots at the Oregon State University Botany and Plant Pathology Research Vineyard.**

| | | | qPCR[c] | | Fisher's exact test (Probability)[d] |
| --- | --- | --- | --- | --- | --- |
| | | | Positive | Negative | |
| Laboratory-qLAMP[a] | 2013 | Positive | 146 (46%) | 13 (4%) | <0.0001[*] |
| | | Negative | 42 (13%) | 115 (37%) | |
| | 2014 | Positive | 36 (16%) | 8 (3%) | <0.0001[*] |
| | | Negative | 61 (27%) | 123 (54%) | |
| Grower-qLAMP[b] | 2014 | Positive | 2 (3%) | 4 (5%) | 0.22[*] |
| | | Negative | 9 (13%) | 58 (79%) | |

Notes:
[a] "Positive" and "Negative" indicate the number of samples for which *E. necator* DNA was detected and not detected, respectively, as tested by L-qLAMP ($n = 316$ in 2013 and $n = 228$ in 2014) assays as described in the text.
[b] G-qLAMP ($n = 73$ in 2014) assessed by growers using mobile qLAMP devices (Diagenetix, Inc., Honolulu, HI, USA) as described in the text.
[c] qPCR results based on TaqMan® probe with minor groove binder for detecting *E. necator* DNA. "Positive" and "Negative" indicate the number of samples for which *E. necator* DNA was detected and not detected, respectively. qPCR detection data based on quantitative data from (*Thiessen, Neill & Mahaffee, 2017*).
[d] Fisher's exact test was used to assess the null hypothesis that each LAMP assay was significantly different from the qPCR assay.
[*] Significant chi-squared test at $P < 0.05$ of qLAMP and qPCR assays.

Regardless of polymerase used, *Bst* or ISO-001, reaction efficiency and sensitivity to *E. necator* DNA was reduced compared to assays conducted prior to implementation of field testing. Different distributers of the Optigene Isothermal Mastermix were also tested to determine if the decreased sensitivity was caused by storage or shipping errors; however, there was no difference among master mix vendors. It was not possible to test previous lots of the master mix prior to the observed decrease in sensitivity. The assimilating probe was removed and gel electrophoresis was used to compare with and without probe presence, and no difference was observed in amplification. There was also no difference between different primer and probe manufacturers, which also suggests there were no differences in manufacturing process.

The concentrations of inhibitor removal compounds within the master mix were assessed, including 2% polyvinylpyrrolidone (PVP) 40, EDTA, and BSA concentrations, to determine if inhibitor presence was causing decreased reaction efficiency, and no differences were observed for inhibitor removal compounds. In addition to testing master mix removal of inhibitors, three DNA extraction methods (extractions with pH 7.5, 10 mM Tris-0.1 mM EDTA buffer (Affymetrix, Santa Clara, CA, USA), PVP 40 (Sigma-Aldrich, St. Louis, MO, USA) in DEPC-treated water, and PowerSoil® DNA extraction kit (Mo Bio Laboratories, Inc., Carlsbad, CA, USA)) were assessed with separate field collected spore samples. No differences were observed in amplification time or efficiency when testing each side-by-side extraction method.

To test the optimal reaction temperature of the polymerase, temperatures between 60 and 70 °C were examined to find the optimal reaction temperature. Lower spore quantities (10 spores or less) amplified at 62 °C. A last effort to determine if the effect

was due to degradation of reagents or primers during the growing season, all reagents, primers, and probe were replaced; however, the decreased sensitivity to *E. necator* DNA was still observed. Despite targeting various portions of the reaction and extraction, the cause for loss of qLAMP assay sensitivity remains undetermined.

## DISCUSSION

A highly sensitive qLAMP assay was successfully developed using a simple DNA extraction method for use by growers or crop consultants to use as a decision aid for timing fungicide applications similar to *Thiessen et al. (2016)* and *Thiessen, Neill & Mahaffee (2017)*. The qLAMP assay developed was sensitive to *E. necator* DNA with one spore amplifying 80% ($n = 10$) using the simplified DNA extraction. This sensitivity indicated that the assay should be suitable to detect inoculum (i.e., ascospores) at low concentrations (<10 spores) and aid management decisions. However, the qLAMP assay consistently underrepresented spore quantities later in the growing season compared to the qPCR assay, which may be due to an increase in the presence of PCR inhibitors (such as pollen, humic acids from soil, spider webs, etc.) found in air samples (*Wilson, 1997*) that may not have been removed by the rapid Chelex DNA extraction. In early DNA extraction testing prior to qLAMP sensitivity loss, the PowerSoil® extracted DNA showed more consistent amplification of field samples than the other extraction methods (*Thiessen et al., 2016*); however, the PowerSoil® DNA extraction kit requires a larger time commitment and several steps that may not be feasible for in-field DNA extractions. The LAMP assay has been widely described as more tolerant to inhibitors than qPCR (*Francois et al., 2011*; *Kaneko et al., 2007*), but it appears that the LAMP assay tolerates different inhibitors than the qPCR assay (*Nixon et al., 2014*). Additionally, the qLAMP $R_T$ variance from one to 10 spore samples (Fig. 2) was so great that they cannot be distinguished. This variance is likely due to using DNA extractions of each spore concentration as opposed a dilution from higher spore concentration as is typically done (*Mahaffee & Stoll, 2016*). Because the LAMP assay is not limited by temperature cycles, annealing is reliant on proximity of DNA to the polymerase and primer set (*Notomi et al., 2000*), and the improved sensitivity with lower annealing temperatures is likely the result of lower specificity of primers rather than optimal reaction temperature. In reactions with lower quantities of DNA (e.g., one and 10 spores), more time may be required for the polymerase, primers, and target DNA to meet, which may explain the variability of $R_T$ values of low spore quantities (Fig. 2). The inhibition of the field qLAMP assay and the difficulty of differentiating low spore quantities indicates that the assay currently has more utility as a qualitative inoculum detection tool as opposed to quantitative assessment of inoculum availability.

The G-qLAMP results were significantly different from the qPCR detection results ($P = 0.22$) (Table 2). This may be due to difficulty in assessing positive detections from the output of the mobile device. The curve smoothing algorithm used by the device application (G-qLAMP) often produced curves that drifted linearly with $R_T$ values reported even though there was no detectable amplification using gel electrophoresis. Growers conducting the q-LAMP assay were directed to ignore curves that ascended
linearly due to curve smoothing; however, this may have caused growers to be overly-conservative in determining positive detections. Additionally, the grower-conducted q-LAMP occurred in 2014 when the loss of q-LAMP sensitivity was observed and there was very low disease.

The L-qLAMP assay detection results were similar to qPCR assay detection results in both 2013 and 2014, but true positive and true negative proportions were variable between years. This variability may be due to the presence of inhibitors. In 2013, the source of stainless-steel rod material was changed from previous testing, and significant inhibition of DNA amplification was observed. After troubleshooting various rod cleaning processes and DNA extraction techniques, a hexane soak was added to the steel rod cleaning protocol to remove oils prior to sterilization and 5% Chelex 100 was used as the extraction buffer. After the hexane wash step addition, the accuracy of samples was improved to 85%, and the misclassification rate was reduced from 17% to 14%. In addition to inhibitors from the rods, the variability of inhibitors from field collections may have caused inconsistencies in qLAMP assay detection results compared to the qPCR assay detection results. Early in the growing season, the weather in the region is characterized by frequent precipitation events that limit pollen and insect flight. Later in the growing season, pollen, insects, birds, and soil particulates are abundant in the air, and subsequently on the sampling rods. RNAses, DNAses, humic acids, and other heavy metals may not be removed when using the chelex DNA extraction (qLAMP template), but are removed during the Powersoil DNA extraction (qPCR template). The results from the qLAMP had lower true positive proportions and true negative proportions than that of turbidimetric LAMP previously developed (*Thiessen et al., 2016*). These reductions may be due to other factors besides amplification inhibitors, such as manufacturer differences, degradation of polymerase, inclusion of probes, or buffering of the qLAMP reaction (*Corless et al., 2000*; *Roux, 2009*).

Using the qLAMP assay for field detection and quantification of fungal pathogens may not be as feasible as previously thought due to the random loss of assay sensitivity and potential inhibition of polymerase activity by environmental contaminants. Redesigning primers was another potential approach to examining the cause of the reduced sensitivity; however, the primer set used here was the result of two previous redesigns during development and testing and there was not sufficient heterogeneity in other regions of the ITS. Additionally, the LAMP assay quantification was also affected by numerous inhibitors, such as soil, pollen, or insect debris, found in field collected samples. LAMP is capable of tolerating some inhibitors that affect PCR assays (*Francois et al., 2011*); however, to determine the extent that LAMP assays are capable of tolerating inhibitors, each potential inhibitor should be tested (*Nixon et al., 2014*). Other LAMP assays developed have utilized more complex DNA extractions to reduce the effect of inhibitors on amplification for quantitation of DNA (*Harper, Ward & Clover, 2010*; *Kubota et al., 2011*; *Mori et al., 2004*); however, complex DNA extraction techniques are likely to preclude field implementation of LAMP assays and increase assay costs. The observed inconsistency indicates that the developed qLAMP assays might not be robust enough for commercial implementation.

The LAMP assay was developed due to reports of high sensitivity and specificity to target DNA, tolerance of the reaction to the presence of reaction inhibitors, and the potential for use by growers or crop consultants using handheld LAMP devices such as the BioRanger (Diagenetix, Inc., Honolulu, HI, USA) or the Genie II and III (Optigene Ltd, West Sussex, UK) (*Kubota et al., 2011*, *2008*; *Mori et al., 2004*, *2001*; *Notomi et al., 2000*; *Temple & Johnson, 2011*; *Tomlinson, Dickinson & Boonham, 2010*); however, field testing of the qLAMP assay for *E. necator* revealed an unidentifiable degradation of the sensitivity of the assay to the target DNA. The qLAMP assay may still be a useful tool for field inoculum detection, but further analysis of the system is required to determine the specific cause of the degradation of the assay.

At the time this research was initiated the LAMP technology was the most advanced for inexpensive field application and thus selected for investigation over other potentially suitable technologies. However, other DNA amplification techniques have since become more accessible for field use (*Marx, 2015*), including qPCR (BioMeme, Inc., Philadelphia, PA, USA), recombinase polymerase amplification (*Piepenburg et al., 2006*), and helicase-dependent isothermal DNA amplification (*Vincent, Xu & Kong, 2004*). These assays require minimal DNA preparation, are capable of real-time data, and may be easily adapted to the air samples used here but require evaluation. There are several reviews that discuss the advantages and disadvantages of these technologies (*Craw & Balachandran, 2012*; *Gill & Ghaemi, 2008*; *Mahaffee, 2014*; *Niemz, Ferguson & Boyle, 2011*; *Yan et al., 2014*).

## CONCLUSION

A highly sensitive qLAMP assay was developed using a simple DNA extraction method for use by growers or crop consultants utilizing inoculum detection; however, the qLAMP assay consistently underrepresented spore quantities later in the growing season compared to the qPCR assay. Additionally, the qLAMP assay lost sensitivity to low spore quantities (<10 spores) in the 2014 sampling period, and the cause was not determined during the course of this study. Grower-conducted inoculum monitoring technologies, like the qLAMP assay developed in this study, may provide an inexpensive tool for producers to apply targeted fungicide applications based on inoculum presence and concentration. Given the limitations described herein, more assessment of the qLAMP assay degradation is necessary before utilizing it as a monitoring tool for *E. necator* inoculum concentrations.

## ACKNOWLEDGEMENTS

We thank the technical support of Andy Albrecht, Cole Provence, Chris Gorman, and Jim Eynard. We also thank anonymous reviewers for their helpful suggestions to improve the manuscript. We especially thank the numerous vineyard managers that collaborated on the project. The use of trade, firm, or corporation names in this publication is for the information and convenience of the reader. Such use does not constitute an official endorsement or approval by the United States Department of Agriculture or the

Agricultural Research Service of any product or service to the exclusion of others that may be suitable.

### Funding

This work was supported by the American Vineyard Foundation, the Oregon Wine Board, and USDA-ARS CRIS 5358-22000-039-00D. The funders had no role in study design, data collection and analysis, decision to publish, or preparation of the manuscript.

### Grant Disclosures

The following grant information was disclosed by the authors:
American Vineyard Foundation.
Oregon Wine Board.
USDA-ARS CRIS: 5358-22000-039-00D.

### Competing Interests

The authors declare that they have no competing interests.

### Author Contributions

- Lindsey D. Thiessen conceived and designed the experiments, performed the experiments, analyzed the data, contributed reagents/materials/analysis tools, prepared figures and/or tables, authored or reviewed drafts of the paper, approved the final draft.
- Tara M. Neill conceived and designed the experiments, performed the experiments, contributed reagents/materials/analysis tools, authored or reviewed drafts of the paper, approved the final draft.
- Walter F. Mahaffee conceived and designed the experiments, authored or reviewed drafts of the paper, approved the final draft.

### Data Availability

The raw CT values and positive negative values of detection events, as well as standard curve development and sensitivity of assay, are provided in Supplemental Dataset Files.

### Supplemental Information

Supplemental information for this article can be found online at http://dx.doi.org/10.7717/peerj.4639#supplemental-information.

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
