# Peer review of "Development of a quantitative loop-mediated isothermal amplification assay for the field detection of Erysiphe necator"

_PeerJ, doi:10.7717/peerj.4639_

## Round 0.1 · original submission · Minor Revisions

Reviewers have provided comments that you are requested to revise. I would particularly emphasize the optimization of the DNA extraction protocol and the comparison with other methods. Please see below.

Reviewer 1 ·

Basic reporting

No comments

Experimental design

It is not clear how the samples are treated in the field (for testing in the LAMP assay). If DNA extraction, i.e., spore disruption, needs lab equipment, doesn’t that defeat the purpose? This is not explained well; the M&M section only refers to the DNA isolation in a lab setting.

The authors state that the LAMP reaction in this study is modified from a previously described report (Thiessen et al Plant Pathol 2016, 65,238-249), some of there parameters of the LAMP of the two systems is different, did the authors assess the optimization of the LAMP protocol?


The authors state that the reason for developing this protocol has something to do with a need for detection of Erysiphe necator to combat the disease by monitoring the inoculum. However, the authors didn't provide a solution for the detction of E. necator at early stage of infection or in grape plants that develop powdery mildew symptoms. Will early detection provided by the proposed protocol allow more rapid screens for resistance? Will it enable the timely spraying of fungicide?

Validity of the findings

No comment

Additional comments

This is a fairly well-written manuscript that presents useful information, however, there is not a good justification for the work which essentially represents the development of a LAMP detection systme that feasible for monitoring Erysiphe necator in its infection cycles thus guiding fungicide spraying.

Reviewer 2 ·

Basic reporting

The basic reporting of this manuscript is appropriate, I especially like the supplemental dataset that was applied.

Experimental design

The experimental design is appropriate, I think involving growers in this applied technology is really a great validation of the technology.

Validity of the findings

The findings are valid and this publication should have an impact by involving the end user. I feel the data is robust.

Additional comments

Review of PeerJ manuscript 21733v1

This manuscript titled “Development of a quantitative loop-mediated isothermal amplification assay for the field detection of Erysiphe necator” looks novel diagnostic tools for GPM. The manuscript is interesting an appropriate for PeerJ and is very well written, however, I have some comments I wish the author would consider prior to publication.

General Comments:

1) My biggest comment on this manuscript is the lack of DNA concentration, most researchers working with this technology will not be able to count conidia and it would be helpful to have an actual DNA concentration (either using Qbit or Nanodrop). I think it would be helpful to others that want to utilize this approach to detect/quantify E. nector. This could also be added to a footnote in Figure 1. It doesn’t need to be extensive, I would just like to know how much DNA is in 1,000 spores (I realize this is a complicated system because of the possible contamination with host DNA). I would add this to the results section also (around line 183, as a bonus it might make this section a bit longer).

2) Please discuss other isothermal approaches in the Discussion, this section of the manuscript is pretty short. Discussions of Helicase Dependent Amplification (HDA) and Recombinase Polymerase Amplification (RPA) would be helpful since they seem to be the “competing technologies” in the area. Why is the LAMP approach the most appropriate/easy to use? Is it cheaper? Better machines etc. I would add this to around

3) Lines 102-108: Can you please place primer/probe sequences into a Table, it is really difficult when working with diagnostic papers if they are in text due to formatting problems. I think many researchers will be interested in the FRET based LAMP probe you used and how it was designed in a visual format.

4) Line 110: I am not sure if CT is really appropriate, I found LAMP literature in reviewing this that just used “Reaction time (minutes)”. I feel like this term CT may be used incorrectly throughout the literature, it really isn’t a cycle, but I agree it is a threshold.

5) Table 1: Spell out “Laboratory” for L and “Grower” for G in the table itself. I realize it is in the caption, but I ran to this table first, then I had to hunt for it a bit.

6) Figure 1: Can you provide a line for this figure and potentially some error/std deviation bars?

7) Figure 2: Is CT value the appropriate name since it isn’t really a “cycle” Other isothermal literature actually counts the minutes for amplification.

8) Figure 3: Is there anyway to log transform this data? the LAMP and qPCR appear pretty different in this figure but I wonder if it is one particular sample/possible contamination? (I have heard it can be a problem with this technology). I suggest at 7/16 for Part A and 5/27 for Part B that you provide the application for those specific samples in the text to highlight the inconsistency a bit more in the text (around line 197).

9) Line 240: Some of this would be better talked about in the discussion section in discussing the optimization of temperature.

---

## Round 0.2 · accepted · Accept

The comments have been addressed as suggested by the reviewers

# Reviewer 2 ·

Basic reporting

The authors have addressed many of my suggested changes and I recommend publication.

Experimental design

No further comments on experimental design.

Validity of the findings

I find their findings valid and worthy of publication.

Additional comments

This manuscript has addressed many of my comments in the original submission. I recommend publication.